# Demolition Waste Potential for Completely Cement-Free Binders

**DOI:** 10.3390/ma15176018

**Published:** 2022-08-31

**Authors:** Ahmed Anees Alani, Ruslan Lesovik, Valery Lesovik, Roman Fediuk, Sergey Klyuev, Mugahed Amran, Mujahid Ali, Afonso R. G. de Azevedo, Nikolai Ivanovich Vatin

**Affiliations:** 1University of Anbar, Ramadi 31001, Iraq; 2Department of Building Materials Science, Products and Structures, Belgorod State Technological University Named after V.G. Shukhov, 308012 Belgorod, Russia; 3Central Research and Design Institute of the Ministry of Construction, Housing and Utilities of the Russian Federation, 119331 Moscow, Russia; 4Polytechnical Institute, Far Eastern Federal University, 690922 Vladivostok, Russia; 5Peter the Great St. Petersburg Polytechnic University, 195251 St. Petersburg, Russia; 6Department of Civil Engineering, College of Engineering, Prince Sattam Bin Abdulaziz University, Alkharj 16273, Saudi Arabia; 7Department of Civil Engineering, Faculty of Engineering and IT, Amran University, Amran 9677, Yemen; 8Department of Civil and Environmental Engineering, Universiti Teknologi PETRONAS, Seri Iskandar 32610, Malaysia; 9LECIV—Civil Engineering Laboratory, UENF—State University of the Northern Rio de Janeiro, 2000 Av. Alberto Lamego, Campos dos Goytacazes, Rio de Janeiro 28013-602, Brazil

**Keywords:** cement-free binders, concrete waste, clinker minerals, microstructural studies

## Abstract

Due to renovation and fighting in the world, a huge accumulation of construction and demolition waste is formed. These materials are effectively used as aggregates, but there is very little information about the use of scrap concrete to create cementless binders. The purpose of the work is to be a comprehensive study of the composition and properties of concrete wastes of various fractions with the aim of their rational use as cementless binders. The scientific novelty lies in the fact that the nature of the processes of structure formation of a cementless binder based on sandy fractions of the screening of fragments of destroyed buildings and structures, as a complex polyfunctional system, has been theoretically substantiated and experimentally confirmed. Different percentages of non-hydrated clinker minerals in concrete scrap were determined. In the smallest fraction (less than 0.16 mm), more than 20% of alite and belite are present. Waste of the old cement paste is more susceptible to crushing compared to the large aggregate embedded in it, therefore, particles of the old cement paste and fine aggregate predominate in the finer fractions of the waste. Comprehensive microstructural studies have been carried out on the possibility of using concrete scrap as a completely cementless binder using scanning electron microscopy, X-ray diffraction analysis, and differential thermal analysis. It has been established that for cementless samples prepared from the smallest fractions (less than 0.315 mm), the compressive strength is 1.5–2 times higher than for samples from larger fractions. This is due to the increased content of clinker minerals in their composition. The compressive strength of the cementless binder after 28 days (7.8 MPa), as well as the early compressive strength at the age of 1 day after steaming (5.9 MPa), make it possible to effectively use these materials for enclosing building structures.

## 1. Introduction

To date, a number of sustainable cementitious materials have been developed, representing promising ways to produce green concrete. For example, alkali-activated materials (or sometimes called geopolymers) are one solution. Xiao et al. [1] analytical investigated of phase assemblages of alkali-activated materials in CaO-SiO_2_-Al_2_O_3_ systems using the management of reaction products and designing of precursors. Manjunatha et al. [2] researched engineering properties and environmental impact assessment of green concrete prepared with PVC waste powder as a step towards sustainable approach. Murthi et al. [3] developed of green concrete using effective utilization of autoclaved aerated concrete brick trash as lightweight aggregate. Chen et al. [4] studied green and sustainable concrete in integrating with different urban wastes.

One of the key problems in the world is the large accumulation of construction waste generated as a result of natural disasters due climate change, and man-made calamities or the demolition of old buildings [5]. In particular, in Iraq, about five billion tons of concrete and reinforced concrete waste have accumulated [6,7,8]. In the near future, after the final dismantling of substandard buildings and structures, the volume of concrete scrap will reach 7–8 billion tons [9]. In addition, a large amount of man-made concrete wastes has been accumulated on the territory of many countries [10]. In this regard, the utilization of concrete scrap from destroyed buildings and structures for the production of building composites seems promising.

Research on the recycling of old concrete as a raw material for the manufacture of new products and structures is carried out in many countries [11,12,13,14,15,16,17,18,19,20,21,22]. At the same time, the main use of recycled concrete is aimed at the manufacture of aggregates [23,24,25,26,27,28]. Nasr et al. [29] found that when concrete is crushed, a thin layer of the hydrated phase remains on the grains of the reused aggregate. This provides an increase in adhesion between the fresh cement matrix and aggregate grains of different fractions and nature, which can be arranged in a row according to the strength of the solution: quartz < limestone < clinker. Due to this, products prepared using concrete scrap improved physical and mechanical properties, including crack resistance and resistance to dynamic loads [30,31,32,33,34].

Ilcan et al. [35] applied waste from the demolition of buildings and structures for the manufacture of geopolymer composites with good rheological properties. Hou et al. [36] used the powder fraction of concrete scrap to create low-cement materials. Chernysheva et al. [37] improved of composite performances using recycled concrete aggregates. Fediuk et al. [38,39,40,41,42] increased recycled concrete binder activity using mechanoactivation technique. Amran et al. [43] reviewed fly ash-based eco-efficient concretes with recycled demolition waste [44]. Volodchenko et al. [45] researched non-autoclaved lime wall materials production using and concrete scrap.

Because of a comprehensive analysis of the literature in recent years [43,46,47,48,49,50,51,52,53,54], a few data on the use of concrete scrap for create cement-less binders were established. When deciding on the rational use of concrete scrap, it is necessary to take into account its service life, the conditions for gaining basic strength, and the content of coarse or fine aggregate [55,56,57,58,59]. Non-hydrated clinker particles may be present in the cement matrix of concrete, which, upon repeated grinding and hydration, form a new hardening structure [60,61,62,63]. Housing stock in Ramadi (Iraq) 1980–2018 constructions are mainly represented by houses made of prefabricated reinforced concrete [64,65]. Concrete and reinforced concrete products are not subjected to heat and moisture treatment during the manufacturing process, but harden under a waterproofing film to a grade strength of 28 days in atmospheric conditions (t = 35–50 °C) due to the hot dry climate of Iraq [66]. During the operation of products, the cement paste is no longer subjected to additional hydration due to the lack of the required amount of moisture [67,68].

As is known, by day 28, complete hydration is observed in C_3_A and C_4_AF, while C_3_S is hydrated by 70–75%, and C_2_S by 45–50%, respectively, the amount of non-hydrated C_3_S can reach 25%, and C_2_S—40–50% [69,70]. Therefore, concrete scrap from destroyed buildings and structures, in addition to aggregates and C-S-H-phase, as a rule, contains clinker particles capable of further hardening [71].

Modern achievements in building materials science declare that only by controlling the processes of structure formation in the hardening system can it be possible to obtain a binder based on technogenic raw materials from destroyed buildings and structures with improved performance properties and a given microstructure [72,73,74,75,76,77,78]. To do this, it is necessary to control the production technology at all its stages, such as the development of optimal compositions, the use of various methods of raw material activation, the use of chemical additives, and a number of other methods [79]. However, the goal of the article is to study the composition and properties of concrete waste of various fractions in order to rationally use them as a cement-less binder. The tasks to achieve the goal of the article were: to clarify the advantages of recycling the fine fraction of crushed concrete scrap screenings as a binder as an active component of concrete; study of the crushability of recycled concrete components; determination of the content of non-hydrated clinker minerals in various fractions of crushed concrete scrap; studies of phase transformations in fractions of concrete scrap; assessment of the potential for hydraulic hardening of the resulting demolition waste fractions.

The scientific novelty lies in the fact that the nature of the processes of structure formation of a cementless binder based on sandy fractions of the screening of fragments of destroyed buildings and structures, as a complex polyfunctional system, has been theoretically substantiated and experimentally confirmed.

## 2. Materials and Methods

### 2.1. Materials

For the research, concrete scrap of destroyed buildings and structures of the cities of Al-Anbar province (Iraq) was used. The chemical composition of concrete scrap is given in Table 1.

The concrete scrap was milled by a laboratory jaw crusher and dispersed into fractions of 0.00–0.16 mm, 0.16–0.315 mm, 0.315–0.63 mm, 0.63–1.25 mm, 1.25–2.5 mm, and 2.5–5 mm in accordance with the Russian standard GOST 8735-88 (Table 2).

### 2.2. Experimental Methodology

Experimental research of materials at all stages was carried out using standard methods of X-ray diffraction (XRD) analysis by a Bruker AXS D8 Advance powder X-ray diffractometer. Study of the morphological features of the microstructure was carried out by a MIRA3 (Tescan, Brno, Czech) scanning electron microscope (SEM) with the ability to conduct energy dispersive spectroscopy (EDS).

Thermal studies (derivative thermal analysis and thermogravimetry) were carried out by a DTG-60H thermogravimetric analyzer (Shimadzu, Kyoto, Japan). A PSH-10a device (Pribory Khodakova, Moscow, Russia) carried out assessment of the specific surface area of materials after grinding. An Analysette 22 diffraction particle size analyzer (NanoTec, Feldkirchen, Germany) studied particle sizes analysis of the powders.

To study the potential for hydraulic hardening, the powders were ground in a laboratory mill to a specific surface area of 400 m^2^/kg. The powders obtained by grinding individual fractions were mixed with water until the paste of normal density was obtained, and sample cubes 700 mm × 700 mm × 700 mm in size were molded. Normal density slightly decreased when moving from the smallest fractions (water–binder ratio = 0.32) to coarse ones (water–binder ratio = 0.3). The compressive strength of the binders was determined according to the Russian standard GOST 10180-2012. The loading rate of the samples was uniform throughout the entire test time at the level of 7 kN/min. It was ensured that the deviations from the dimensions along the length of the ribs of the samples did not exceed 1%. To obtain adequate results, six samples were made in each series. Compaction of the binder mixture in a mold was carried out by vibration. Part of the fabricated samples hardened for 2, 7, and 28 days under normal conditions, and another part fabricated one day during steaming according to the “3 h + 8 h + cooling” mode (isothermal holding temperature 80 °C).

## 3. Results and Discussion

Below is a step-by-step analysis of the studies carried out in accordance with modern requirements [80,81,82].

### 3.1. XRD Patterns

The advantages of reusing the fine fraction of screenings from crushing concrete scrap as a binder were revealed with the help of X-ray diffraction analysis (Figure 1). It was ensured that the deviations from the dimensions along the length of the ribs of the samples did not exceed 1%. To obtain adequate results, six samples were made in each series. Compaction of the binder mixture in a mold was carried out by vibration. Cement paste minerals are mainly represented by alite (d = 2.77; 2.608; 2.606… Å) and belite (d = 2.75; 2.71… Å) clinker minerals, partially crystallized calcium silicate hydrates (d = 9.8; 4.9; 3.07; 2.85; 2.80; 2.40; 2.00; 1.83… Å); and portlandite (d = 4.93; 3.11; 2.63; 1.93; 1.79; 1.69… Å). At the same time, most of the cement layers, firmly fixed on the surface of the aggregate of quartz sand and gravel during the previous exploitation and repeated crushing, is carbonized, in addition, quartz is present in the concrete scrap.

On the XRD patterns of the milled concrete scrap, there are peaks characteristic of coarse and fine aggregates such quartz (4.26; 3.34; 2.46; 1.82; 1.54… Å); feldspars (microcline, albite) (3.24; 3.19… Å), and biotite (10.069… Å). An analysis of X-ray patterns of concrete scrap by fractions showed that the intensity of C_3_S and C_2_S reflections reduces with the transition from a fraction of 0.00–0.16 mm to a fraction of 2.5–5 mm (Figure 2).

At the same time, the amount of quartz and minerals that are characteristic of coarse aggregate increases. The finest fractions of concrete scrap (up to 0.16–0.316 mm) contain the maximum amount of alite and belite compared to larger fractions that can harden when interacting with water.

### 3.2. Microscopy

The different composition of the investigated fractions is due to the various crushability of the recycled concrete components. For the coarse aggregate made of gravel, the crushing strength is higher than that of the mortar part of old concrete, therefore, gravel grains are crushed worse, and mainly accumulate in coarse fractions of crushed concrete scrap. Cement paste is more susceptible to crushing, and therefore, particles of the cement matrix and quartz sand predominate in finer fractions (Figure 3).

### 3.3. DTG Patterns

In order to confirm the foregoing, thermal analysis methods were used to study phase transformations in various fractions of the concrete scrap. The results showed that the processes for all fractions have five pronounced endothermic effects (Figure 4). The first endoeffect manifests itself at temperatures from 88 to 102 °C and is due to the loss of adsorbed moisture.

The second endothermic effect is recorded in the temperature range of 439–449 °C. The beginning of the endothermic effect at these temperatures, which can be seen in the DTA patterns of the samples, is associated with the dehydration of Ca(OH)_2_, which occurs as a result of the hydration of the C_3_S phase.

The third and fourth endoeffects at temperatures of 574 and 689 °C correspond to the dehydration of portlandite and its transition to CaO. Lime released during cement hydration absorbs CO_2_ from the atmosphere to form CaCO_3_. An endothermic effect above 750 °C indicates the presence of CaCO_3_. The intensity of this peak increases with increasing hydration time.

The next endothermic effect and significant weight loss are observed in the temperature range from 696 to 717 °C, with the formation of C_2_S. The last peak and weight loss in the temperature range from 900 to 940 °C correspond to the release of structural water from muscovite.

### 3.4. Physical and Mechanical Tests

Further, the results of physical and mechanical tests are shown in Table 3.

The highest hydraulic activity was shown by powders of two fractions 0.00–0.16 and 0.16–0.315 mm, which cured both under normal conditions and during steaming. The compressive strength of these samples is 50–100% higher than that of ones prepared from powders of larger fractions. The powder from the pulverized fraction showed the best result in terms of strength. This is due to the increased content of clinker minerals in the composition of small fractions of concrete scrap, due to which the hydration process proceeds more intensively, which leads to an increase in mechanical strength in comparison with large fractions and is confirmed by the results of mechanical tests.

It follows from the above that the maximum amount of alite and belite capable of hydration hardening is present in the smallest fractions of 0.00–0.16 mm and 0.16–0.315 mm. With an increase in the size of the fractions, the amount of alite and belite decreases and, at the same time, the amount of quartz and minerals characteristic of coarse aggregate increases.

It was established that for samples prepared from powders obtained by grinding the smallest fractions (0.00–0.16 mm and 0.16–0.315 mm), the compressive strength is 1.5–2 times higher than for samples from larger fractions, which is due to the increased content of C_3_S and C_2_S in their composition. Therefore, for the manufacture of a binder from concrete scrap, it is advisable to use the smallest part of the crushing products with a diameter of up to 0.315 mm.

The limitations of this manuscript lie in the need to minimize the dimensionality of shredded a construction waste. At the same time, the construction waste refers to specially selected concrete components. Relatively low mechanical characteristics determine the scope of the developed binder for enclosing structures of buildings and structures.

## 4. Conclusions

The compositions and properties of concrete wastes of various fractions were studied with the aim of their rational use as a cementless binder. As a result, the following conclusions, possessing the scientific novelty, are slanted:-The advantages of recycling the fine fraction of crushed concrete scrap screenings as a binder were revealed. After concrete crushing, layers remain on the aggregate grains in the form of a mortar component or thin layers of hydrated phases, and there are also finely dispersed particles of cement paste.-The different composition of the investigated fractions is due to the different crushability of the components of recycled concrete. In a coarse aggregate made of gravel, the crushing strength is higher than that of the mortar part of concrete, therefore, gravel grains are crushed worse, and mainly accumulate in large fractions of crushed concrete scrap. Cement paste is more susceptible to crushing, and therefore, particles of the cement matrix and quartz sand predominate in fractions of a smaller size.-A greater amount of cement paste in secondary concrete corresponds to a greater content of non-hydrated clinker minerals alite and belite, which is confirmed by X-ray diffraction analysis data.-The presence of clinker minerals in recycled concrete is also confirmed by differential thermal studies of phase transformations in concrete scrap fractions.-To assess the ability to hydraulic hardening, the obtained fractions of concrete scrap were molded to cubic samples, hardened for 28 days, and tested for compressive strength. The highest hydraulic activity was shown by powders of two fractions 0.00–0.16 and 0.16–0.315 mm, which hardened both under normal conditions and during steaming. The compressive strength of these samples is 50–100% higher than that of ones prepared from powders of larger fractions.

**Prospects for further research** are aimed at the possibility of modifying scrap concrete, expanding the reserves of substandard raw materials, detailing, and deepening research in the direction of studying the features of the processes of structure formation of various cement composites.

## Figures and Tables

**Figure 1 materials-15-06018-f001:**
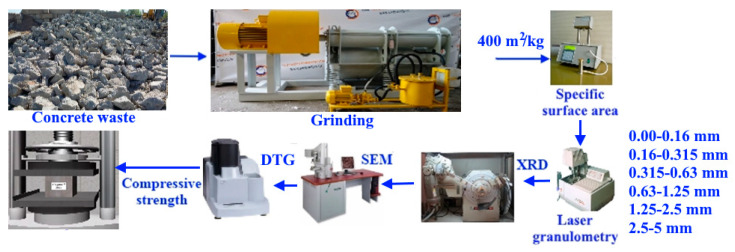
Flow chart of the procedure.

**Figure 2 materials-15-06018-f002:**
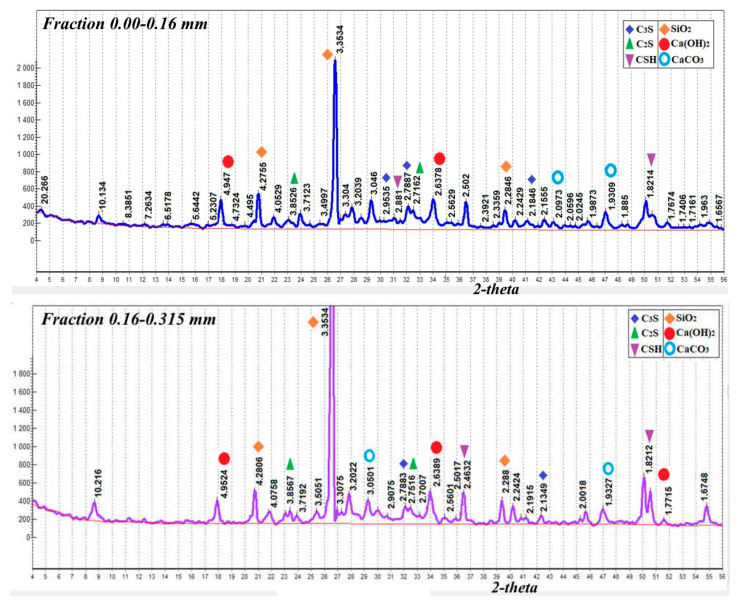
XRD patterns of screening fractions of the concrete scrap crushing. Different colored lines indicate the results for different fractions, and the red line at the bottom is drawn for ease of reference.

**Figure 3 materials-15-06018-f003:**
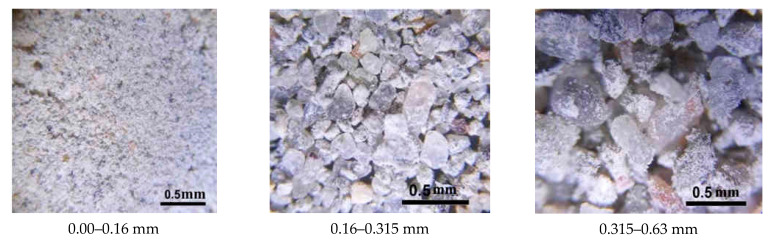
Particles of concrete scrap of various fractions.

**Figure 4 materials-15-06018-f004:**
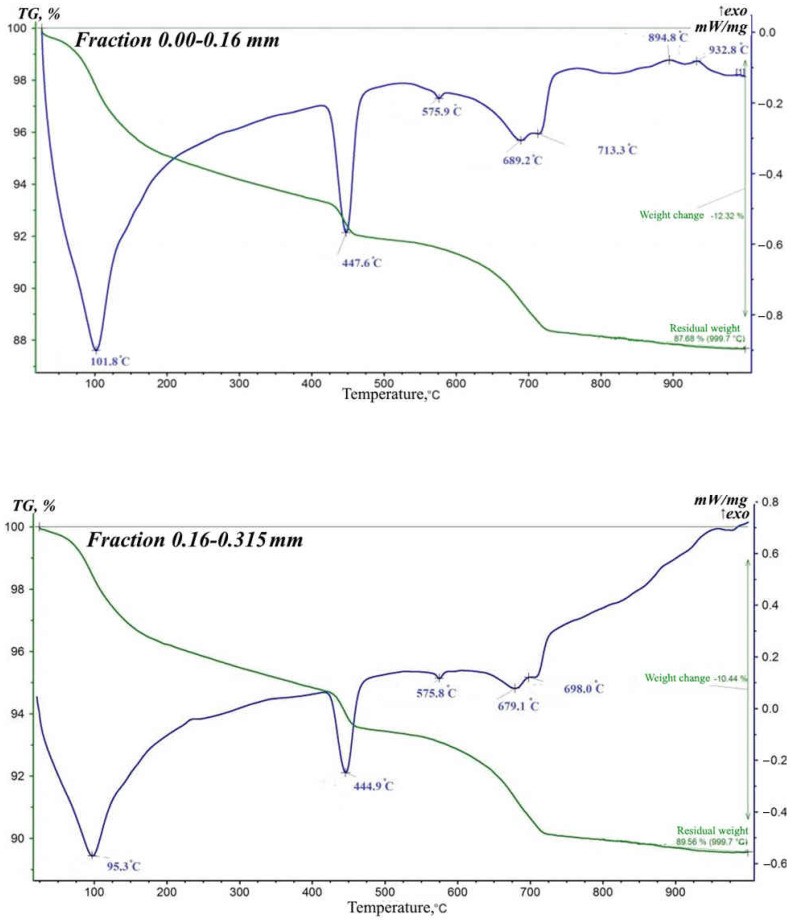
Differential thermal analysis of various fractions of concrete scrap crushing screenings.

**Table 1 materials-15-06018-t001:** Chemical composition of the concrete scrap, %.

Oxide	Value	
	Nominal	Deviation
CaO	30.00	±0.80
SiO_2_	49.50	±0.45
Al_2_O_3_	9.80	±0.20
Fe_2_O_3_	3.20	±0.20
MgO	2.70	±0.06
SO_3_	1.47	±0.10
Na_2_O	1.58	±0.10
K_2_O	1.70	±0.10
other	0.05	-

**Table 2 materials-15-06018-t002:** Particle sizes of the crushed concrete scrap.

Properties	Size of Sieve Openings, mm					Pass Through a Sieve No. 0.16
	2.5	1.25	0.63	0.315	0.16	
Residues on sieves, g:	350	83	113	133	155	166
-partial, %	35	8.3	11.3	13.3	45.5	16.6
-full, %	35	43.3	54.6	67.9	83.4	100

**Table 3 materials-15-06018-t003:** Physical and mechanical properties of binders depending on the fraction.

Fraction, mm	Property				
	Water–Binder Ratio	Compressive Strength, MPa			
		2 Days	7 Days	28 Days	1 Day (after Steaming)
0.00–0.16	0.32	3.2	4.3	7.8	5.9
0.16–0.315	0.32	3.5	3.7	6	4.1
0.315–0.63	0.31	1.7	3	3.8	2.9
0.63–1.25	0.31	2	3.1	3.7	2.7
1.25–2.5	0.31	2.2	2.3	4.4	3.5
2.5–5	0.3	2.1	2	4.1	3.2

## Data Availability

Not applicable.

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
