# Peer review of "Demolition Waste Potential for Completely Cement-Free Binders"

_materials, 2022, doi:10.3390/ma15176018_

Round 1

Reviewer 1 Report

1 The abstract should be revised. The significances in engineering field should be highlighted. 2The authors are suggested to explain the novelty of the paper. 3 In Section 2, the authors are suggested to add a diagram or picture of the flow chart of the procedure. 4 For the analysis in Section 3, further step analysis should be added. The authors are suggested to add comments as well as the references below. They are closely related with the present research. Mechanical Systems and Signal Processing, 2023, 182: 109349  5 For the conclusions, it is a little long. Several brief points of conclusions are enough. The authors are suggested to rewrite the conclusions. 6 The English should be improved considerately.

Author Response

Dear Reviewer 1!

Thank you for your interest in our manuscript. Your valuable comments helped make the manuscript even better. All corrections in the manuscript are highlighted in blue.

Comment 1: The abstract should be revised. The significances in engineering field should be highlighted.

Response: The abstract has been revised

Comment 2: The authors are suggested to explain the novelty of the paper.

Response: Added: «The scientific novelty lies in the fact that the nature of the processes of structure formation of a cementless binder based on sandy fractions of the screening of fragments of destroyed buildings and structures, as a complex polyfunctional system, has been theoretically substantiated and experimentally confirmed»

Comment 3: In Section 2, the authors are suggested to add a diagram or picture of the flow chart of the procedure.

Response: Fig. 1 added

Comment 4: For the analysis in Section 3, further step analysis should be added. The authors are suggested to add comments as well as the references below. They are closely related with the present research. Mechanical Systems and Signal Processing, 2023, 182: 109349

Response: The article presents a step-by-step analysis of the studies carried out in accordance with modern requirements. Subsections have been added to section 3. The article kindly offered by you has been cited

Comment 5: For the conclusions, it is a little long. Several brief points of conclusions are enough. The authors are suggested to rewrite the conclusions.

Response:  Conclusions have been rewritten

Comment 6: The English should be improved considerately

Response: The article has been carefully proofread by a native English speaker

Reviewer 2 Report

In this paper, the cementitious properties of recycled concrete powder were systematically investigated. It was found that the compressive strength of the cementless binder after 28 days (7.8 MPa), as well as the early compressive strength at the age of 1 day after steaming (5.9 MPa), make it possible to effectively use these materials for non-bearing building structures. In general, this is a very interesting and meaningful research. The experiments are well designed and the conclusions are well supported by the results. I have some suggestions:

1. The literature review is not enough. The authors may give a bigger picture of sustainable cementitious materials. Other promising ways of making green concrete should be briefly reviewed. For example, the alkali-activated materials (or sometimes called geopolymer) are one of the solutions. The recently published papers should be reviewed. (e.g., "Analytical investigation of phase assemblages of alkali-activated materials in CaO-SiO2-Al2O3 systems: The management of reaction products and designing of precursors. Materials & Design194, p.108975.").

2. The information of how you made samples are missing or need to be clarified. For example, what is the water-to-cement ratio?

3. The test method of compressive strength should be specified. What is the loading rate?

4. Why do you think your samples can represent most recycled concrete? Will the age of concrete, original mix design influence the results? The authors may give a more indepth discussion on how this technology should be applied in industry and what obstacles people will meet.

5. The limitation of this study should be discussed.

Author Response

Dear Reviewer 2!

Thank you for your interest in our manuscript. Your valuable comments helped make the manuscript even better. All corrections in the manuscript are highlighted in blue.

Comment 1: The literature review is not enough. The authors may give a bigger picture of sustainable cementitious materials. Other promising ways of making green concrete should be briefly reviewed. For example, the alkali-activated materials (or sometimes called geopolymer) are one of the solutions. The recently published papers should be reviewed. (e.g., "Analytical investigation of phase assemblages of alkali-activated materials in CaO-SiO2-Al2O3 systems: The management of reaction products and designing of precursors. Materials & Design, 194, p.108975.").

Response: The first paragraph of the introduction has been added about this. Including the article you kindly recommended has been carefully studied and cited

Comment 2: The information of how you made samples are missing or need to be clarified. For example, what is the water-to-cement ratio?

Response: The water-binding ratio of all samples was demonstrated in Table 3. Part of the fabricated samples hardened for 2, 7 and 28 days under normal conditions, and another part fabricated one day during steaming according to the “3 hours + 8 hours + cooling” mode (isothermal holding temperature 80°C).

Comment 3: The test method of compressive strength should be specified. What is the loading rate?

Response: Added: «The loading rate of the samples was uniform throughout the entire test time at the level of 7 kN/min»

Comment 4: Why do you think your samples can represent most recycled concrete? Will the age of concrete, original mix design influence the results? The authors may give a more indepth discussion on how this technology should be applied in industry and what obstacles people will meet.

Response: Added to the end of third section: «The limitations of this manuscript lie in the need to minimize the dimensionality of shredded a construction waste. In the same time the construction waste refers to specially selected concrete components. Relatively low mechanical characteristics determine the scope of the developed binder for enclosing structures of buildings and structures.»

Comment 5: The limitation of this study should be discussed.

Response: Added to the end of third section: «The limitations of this manuscript lie in the need to minimize the dimensionality of shredded a construction waste. In the same time the construction waste refers to specially selected concrete components. Relatively low mechanical characteristics determine the scope of the developed binder for enclosing structures of buildings and structures.»

Reviewer 3 Report

The topic “Demolition waste potential for completely cement-free binders” will definitely be of interest to the readers of the journal. However, the flow and sentence structure in the manuscript can be improved. Literature is not thorough and the reader is not given an adequate background about the topic, therefore adding some more relevant and updated references is strongly recommended. Also, the future work should be recommended based on the study experimentation presented in the manuscript. Quality of figures could be improved and authors probably need to increase the resolution.

Author Response

Dear Reviewer 3!

Thank you for your interest in our manuscript. Your valuable comments helped make the manuscript even better. All corrections in the manuscript are highlighted in blue.

Comment 1: The topic “Demolition waste potential for completely cement-free binders” will definitely be of interest to the readers of the journal. However, the flow and sentence structure in the manuscript can be improved.

Response: The flow and sentence structure in the manuscript have been improved

Comment 2: Literature is not thorough and the reader is not given an adequate background about the topic, therefore adding some more relevant and updated references is strongly recommended.

Response: Literary review has been significantly supplemented

Comment 3: Also, the future work should be recommended based on the study experimentation presented in the manuscript.

Response: In the end of paper: «Prospects for further research are aimed at the possibility of modifying scrap concrete, expanding the reserves of substandard raw materials, detailing and deepening re-search in the direction of studying the features of the processes of structure formation of various cement composites.»

Comment 4: Quality of figures could be improved and authors probably need to increase the resolution.

Response: Quality of the figures have been improved

Round 2

Reviewer 1 Report

1 The abstract should be revised. The significances in engineering field should be highlighted.

2 The authors are suggested to explain the novelty of the paper.

3 In Section 2, the authors are suggested to add a diagram or picture of the flow chart of the procedure.

4 For the analysis in Section 3, further step analysis should be added. The authors are suggested to add comments as well as the references below. They are closely related with the present research. Mechanical Systems and Signal Processing, 2023, 182: 109349,Mechanical Systems and Signal Processing, 2022, 168: 108624. Tribology International, 2021, 164: 107105 .

5 For the conclusions, it is a little long. Several brief points of conclusions are enough. The authors are suggested to rewrite the conclusions.

6 The English should be improved considerately.

Author Response

Dear Reviewer 1!

Thank you for your interest in our manuscript. Your valuable comments helped make the manuscript even better. All corrections in the manuscript are highlighted in blue.

Comment 1: The abstract should be revised. The significances in engineering field should be highlighted.

Response: The abstract has been significantly revised from an engineering point of view. It contains numerical results and areas of practical application.

Comment 2: The authors are suggested to explain the novelty of the paper.

Response: Added: «The scientific novelty lies in the fact that the nature of the processes of structure formation of a cementless binder based on sandy fractions of the screening of fragments of destroyed buildings and structures, as a complex polyfunctional system, has been theoretically substantiated and experimentally confirmed»

Comment 3: In Section 2, the authors are suggested to add a diagram or picture of the flow chart of the procedure.

Response: Fig. 1 such flowchart of the procedure have been added

Comment 4: For the analysis in Section 3, further step analysis should be added. The authors are suggested to add comments as well as the references below. They are closely related with the present research. Mechanical Systems and Signal Processing, 2023, 182: 109349,Mechanical Systems and Signal Processing, 2022, 168: 108624. Tribology International, 2021, 164: 107105.

Response: The article presents a step-by-step analysis of the studies carried out in accordance with modern requirements. Subsections have been added to section 3. All the articles kindly offered by you have been cited

Comment 5: For the conclusions, it is a little long. Several brief points of conclusions are enough. The authors are suggested to rewrite the conclusions.

Response:  Conclusions have been carefully rewritten

Comment 6: The English should be improved considerately

Response: The article has been carefully proofread by a native English speaker

Reviewer 2 Report

This paper has been revised based on the comments.

Author Response

Thank you for appreciating our article.